

# *De novo* assembly and characterization of the Chinese three-keeled pond turtle (*Mauremys reevesii*) transcriptome: presence of longevity-related genes

Huazong Yin, Liuwang Nie, Feifei Zhao, Huaxing Zhou, Haifeng Li, Xianmei Dong, Huanhuan Zhang, Yuqin Wang, Qiong Shi and Jun Li

College of Life Science, Anhui Normal University, Provincial Key Lab of the Conservation and Exploitation Research of Biological Resources in Anhui, Wuhu, Anhui, China

## ABSTRACT

*Mauremys reevesii* (Geoemydidae) is one of the most common and widespread semi-aquatic turtles in East Asia. The unusually long lifespan of some individuals makes this turtle species a potentially useful model organism for studying the molecular basis of longevity. In this study, pooled total RNA extracted from liver, spleen and skeletal-muscle of three adult individuals were sequenced using Illumina Hiseq 2500 platform. A set of telomere-related genes were found in the transcriptome, including *tert*, *tep1*, and six shelterin complex proteins coding genes (*trf1*, *trf2*, *tpp1*, *pot1*, *tin2* and *rap1*). These genes products protect chromosome ends from deterioration and therefore significantly contribute to turtle longevity. The transcriptome data generated in this study provides a comprehensive reference for future molecular studies in the turtle.

## INTRODUCTION

Longevity has always been a trait of great interest to researchers, and numerous studies have been performed on humans and model organisms to better understand its molecular mechanisms, such as the maintenance of telomeric structure. Synthesized by telomerase (*Blackburn, Greider & Szostak, 2006*; *Bodnar et al., 1998*), telomeres are specialized structures at the ends of eukaryotic chromosomes that help to maintain genome integrity by protecting chromosomes from rearrangements or fusion to each other (*McClintock, 1939*; *Muller, 1938*). Introducing telomerase into normal human retinal pigment epithelial cells and foreskin fibroblasts significantly extends the lifespan of the cells (*Bodnar et al., 1998*). Several protein complexes have also been implicated in longevity. For example, the shelterin complex specifically recognizes and binds to telomeric DNA, preventing the chromosome end from being detected as a DNA double-strand break (*Palm & de Lange, 2008*). Research on the naked mole rat (*Heterocephalus glaber*) suggests that shelterin complex-encoding genes are related to the species' longevity (*Kim et al., 2011*). Synthesized telomeres are shaped and safeguarded by the shelterin complex, an essential component

Corresponding author
Liuwang Nie,
lwnie@mail.ahnu.edu.cn

of telomere function (*De Lange, 2005*) that consists of six proteins: TRF1, TRF2, POT1, RAP1, TIN2, and TPP1 (*Bilaud et al., 1997*; *Broccoli et al., 1997*; *Chong et al., 1995*; *De Lange, 2005*; *Houghtaling et al., 2004*; *Kim, Kaminker & Campisi, 1999*; *Li, Oestreich & De Lange, 2000*; *Liu et al., 2004*; *Xin, Liu & Songyang, 2008*; *Ye et al., 2004*; *Zhong et al., 1992*).

Turtles are an ideal model organism for research on the molecular basis of longevity. They are the most morphologically distinct order to have originated from the late Permian and early Triassic period (about 220 million years ago) (*Li et al., 2008*). In addition to their characteristic shell, turtles also have many remarkable physiological traits, such as anoxia (*Lutz, Prentice & Milton, 2003*) and cold tolerance (*Packard et al., 1997*), temperature-determined sex differentiation (*De Souza & Vogt, 1994*; *Mrosovsky, Dutton & Whitmore, 1984*), and of particular note, a long lifespan. Many individuals have been recorded as living more than 100 years (*Gibbons, 1987*; *Shaffer et al., 2013*). For example, the *Geochelone gigantean* known as Marion's tortoise lived for more than 150 years (*Schmidt & Inger, 1957*), and Lonesome George, a *Geochelone nigra*, was reported to have lived more than 100 years.

In addition, previous reports have described many physical characters that are associated with turtle longevity, such as being reproductively active at very advanced ages and negligible functional impairment with age (*Miller, 2001*). Some authors have linked longevity in turtles to enhanced mechanisms of reoxygenation for surviving brain anoxia (*Lutz, Prentice & Milton, 2003*). A study on European freshwater turtles (*Emys orbicularis*) found that telomere length was only shortened negligibly in adults compared to circulating embryonic blood cells (*Girondot & Garcia, 1999*). However, currently, we do not know whether turtle longevity is associated with telomerase activity or the insulin/IGF-1 signaling pathway.

*Mauremys reevesii* (Geoemydidae) is widespread and common in China, the Korean Peninsula, and Japan (*Van Dijk et al., 2014*). Due to its longevity, it has substantial cultural significance in China as an auspicious omen. In this study, total RNA extracted from liver, spleen, and skeletal muscle of three adult females were used to generate a pooled cDNA library, which was subsequently sequenced on an Illumina Hiseq 2500 platform. Next generation sequencing (NGS) technologies have been broadly applied in genome and transcriptome sequencing (*Reis-Filho, 2009*), due to their greater sensitivity, which supplies accurate results that can detect previously unknown and/or rare genes. The two primary aims of our study were as follows: (1) to better understand the molecular mechanism behind the long lifespan of turtles by identifying longevity-associated genes, and (2) to generate transcriptome data as a useful resource for future studies of turtle longevity and other traits.

## MATERIALS & METHODS

### Ethical approval

Procedures involving animals and their care were approved by the Animal Care and Use Committee of Anhui Normal University under approval number #20140111.

### Sample collection and RNA extraction

Three adult female turtles were collected from our plant at Wuhu, Anhui, China. The liver, spleen, and skeletal muscle were collected and dissected. Tissue samples were stored at

−80 °C. Total RNA was extracted from the tissues separately using a Trizol kit (Invitrogen, CA, USA) according to the manufacturer's protocol. Extracted RNA was quantified with Nanodrop (Thermo, CA, USA) and the integrity and size distribution were checked with agarose gel electrophoresis. High-quality RNA from all three tissues was pooled for cDNA synthesis and sequencing.

## cDNA library construction and sequencing

Two micrograms of pooled total RNA was used for cDNA library construction using TruSeq® RNA LT Sample Prep Kit v2 (Illumina, CA, USA) according to the manufacturer's protocol. We then prepared the synthesized cDNA for sequencing library construction by performing end-repair, 3′-end adenylation, as well as adapter-ligation and enrichment. Sequencing was performed using an Illumina Hiseq 2500 platform (*Quail et al., 2008*) at Genergy Bio-technology Co., Ltd. (Shanghai, China).

## Sequence data processing and *de novo* assembly

The raw reads generated by the Illumina sequencer were saved as fastq format files. Adapter sequences were trimmed and low-quality reads were removed from the raw reads using Trim Galore (version 0.3.5) software. FastQC (version 0.10.1) was used to check the quality of pretreated data; reads that achieved a high Phred score (>28) were used for the assembly. We used Trinity (*Grabherr et al., 2011*), with a k value = 25, to perform *de novo* assembly on the trimmed and quality-checked reads. Sequence data were partitioned into many individual de Bruijn graphs, representing the transcriptional complexity at a given gene or locus. Each graph was processed independently to extract full-length splicing isoforms and to tease apart transcripts derived from paralogous genes. The final Trinity output was analyzed. Gene expression level was calculated using RSEM software (version 1.2.3) (*Li & Dewey, 2011*).

## Blast against turtle's reference proteomes

We used Blastx to query all *M. reevesii* transcripts to the proteomes of the green sea turtle, the western painted turtle, and the soft shell turtle, which were downloaded from the GenBank. The e-value cutoff was set to $1 \times e^{-10}$ and the maximum target number was set to 20. The top results were selected as the annotation of the gene (termed as "unique protein"). A Venn diagram of the homologous genes across the three turtle proteomes was generated with VENNY (*Oliveros, 2007*).

## Functional annotation

Sense and component strands of potential protein coding sequences (CDS) were predicted using Transdecoder in Trinity software v2.0.2 package (http://transdecoder.github.io/), based on a Markov model with default parameters. CDS were then translated into amino acid sequences with reference to a standard codon table. We used Blastp to search the Swissprot/Uniprot database (*Balakrishnan et al., 2005*) with our potential protein sequences as queries. The e-value cutoff was set to $1 \times e^{-3}$. A gene name was assigned to each contig based on the top Blastp hit.

Gene ontology (GO) analyses (*Ashburner et al., 2000*) on all predicted protein sequences were conducted using InterProScan, set to default parameters (*Zdobnov & Apweiler, 2001*).

The GO terms associated with each assembled sequence of the turtle transcriptome were then classified into biological processes, molecular functions, and cellular components. InterProScan was also used to predict the functional domains, signal peptides, and other protein characters by blasting to the Conserved Domain Database Interpro.

We then employed the KEGG Automatic Annotation Server (KAAS) with default settings (*Moriya et al., 2007*) to perform a KEGG pathway analysis (*Kanehisa & Goto, 2000*) on each contig. Telomere-related genes were filtered from the annotation results.

## Analysis of candidate longevity-associated genes

Candidate genes associated with turtle longevity were screened from the logfile generated by Blastp. Two shelterin genes (i.e., *pot1* and *tin2*) were not detected in this step. To search the lost but should be existed genes in *M. reevesii* transcriptome, we used released *pot1* (XM_007063192.1) and *tin2* (XM_005290151.2) genes to blastn against the primary assembled *M. reevesii* transcriptome data. And the *pot1* and *tin2* genes were finally screened out that escaped from functional annotation.

The filtered longevity-associated genes were then blasted to the NR database using Blastn, to verify the accuracy of previous annotations. Candidate gene fragments were aligned, overlapping fragments were assembled into one sequence, and separated fragments were concatenated into a single sequence. FPKM values of each fragment were also checked using the results from the RSEM software.

Screened genes were translated into amino acids, using MEGA 6.06 (*Tamura et al., 2013*) and according to a standard codon table. Amino acid sequences were then aligned with published data using the online alignment tool MAFFT (version 7) (http://mafft.cbrc.jp/alignment/software/) (*Katoh et al., 2002*), with default settings. BioEdit 7.2.3 (*Hall, 1999*) was used to display the alignment results.

# RESULTS

## Sequencing and *de novo* assembly

Total RNA extracted from liver, spleen, and skeletal muscle were used to generate a pooled cDNA sample and subsequently sequenced. A total of 160,998,396 paired-end raw reads of 100 bp length were generated and stored in fastq format (GenBank accession number: SRX1469958). We obtained 152,214,434 (94.5%) high-quality reads with an average length of 98.8 bp. The results of *de novo* assembly yielded 459,911 isoforms, which clustered into 230,085 transcripts with average length of 660 bp and median length of 342 bp. 190 transcripts encompassed over 100 isoforms and 6,051 transcripts encompassed over 10 isoforms with an average of two isoforms per transcript overall. Additionally, the highest number of isoforms found in one transcript was 1,048. These results may indicate the widespread existence of alternative splicing in *M. reevesii*. GC content for the entire final assembly and all protein coding sequences were 46.67% and 51.16%, respectively. Repetitive elements and microsatellites were also analyzed (File S1).

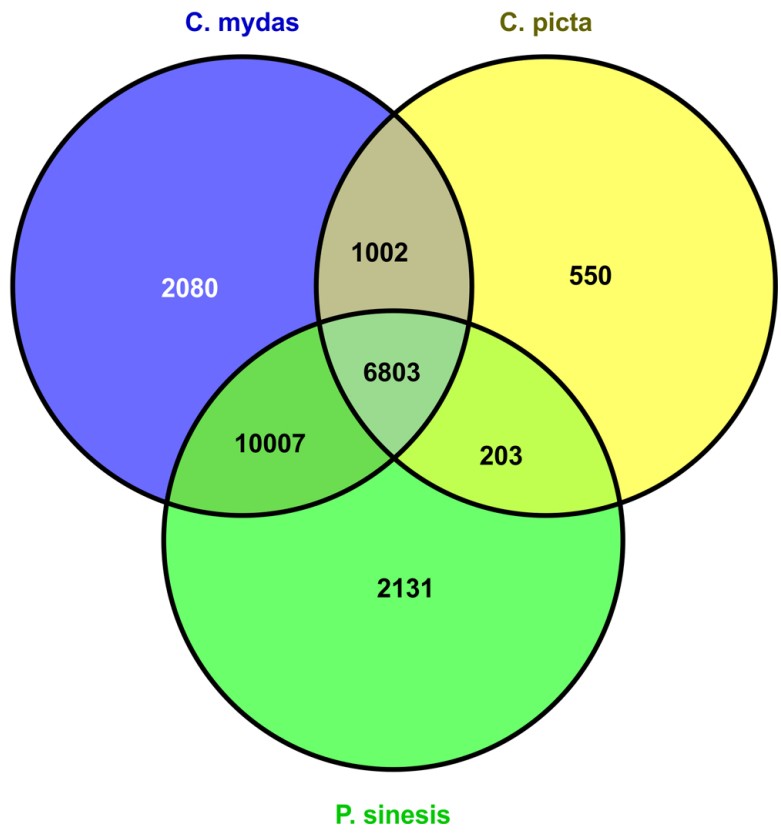

**C. mydas**

**C. picta**

**P. sinesis**

**Figure 1** *M. reevesii* homologous gene detection in diverse turtle proteomes.

**Table 1** Summary of Blastx search results of *M. reevesii* transcriptome.

| Database | Unigene hits | Unique protein |
|---|---|---|
| *Chelonia mydas* | 19,892 | 18,263 |
| *Pelodiscus sinensis* | 19,144 | 27,267 |
| *Chrysemys picta bellii* | 8,558 | 27,198 |

## Comparison with turtle's reference proteomes

The assembly quality of the *M. reevesii* transcriptome was assessed with a Blastx comparison to the reference proteomes of three turtles: the green sea turtle (*Chelonia mydas*), the Chinese soft-shell turtle (*Pelodiscus sinensis*), and the western painted turtle (*Chrysemys picta bellii*) (Fig. 1 and Table 1).x *Chelonia mydas* exhibited the highest degree of similarity. The comparison to *P. sinensis* and *C. picta* yielded slightly more unique proteins than transcripts hits, but overall, we obtained a total of 22,776 positive transcripts hits.

## Transcripts expression level

The expression of each transcript was quantified using RSEM software, set to default parameters. Statistical results are shown in Table 2. The transcripts with the highest levels of expression were related to metabolism and translation activity.

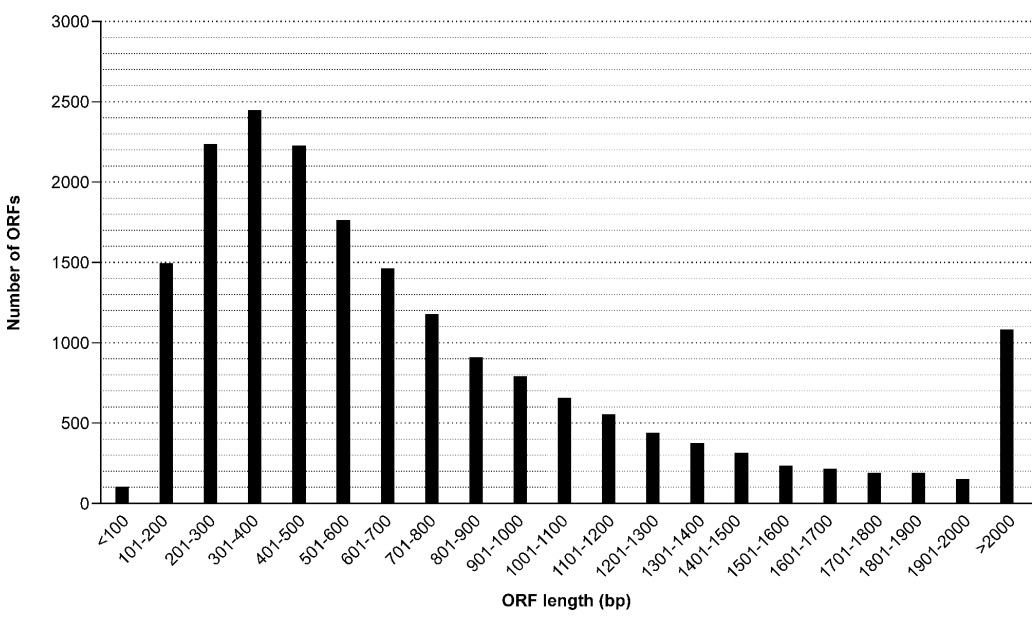

**Figure 2** Length distribution of identified ORF (open reading frames) from the *M. reevesii* transcriptome assembly.

**Table 2** Statistics of FPKM distribution of assembled unigenes.

|  | Unigenes |
|---|---|
| >1,000 FPKM | 101 |
| >100 FPKM | 809 |
| <10 FPKM | 218,717 (95.1%) |
| Max FPKM | 12367.8 |
| Min FPKM | 0 |
| Total | 230,085 |

## Functional annotation

The results of our functional annotation revealed 42,918 (18.7%) transcripts that were predicted to potentially code for proteins. These were then translated into amino acid sequences by referring to a standard codon table. After a blastp search on the resultant amino acid sequences, we obtained a total of 534,077 hits, the best of which corresponded to 18,846 unique protein accessions in the Swissprot/Uniprot database. Length distribution for all opening reading frames ranged from 40 bp to 34,967 bp, with an average length of 874 bp (Fig. 2). The Blastp top-hit species distribution of gene annotations showed the highest homology to *Chelonia mydas* (6,654 annotation results) and *Pelodiscus sinensis* (5,747 annotation results) (Fig. S1). Those two species supplied 65.8% of all annotation information due to their close phylogenetic relationships to *M. reevesii*.

**Table 3  Statistics of predicted protein domains and site characters.**

| Domains | Counts |
| --- | --- |
| Zinc finger, C2H2-like | 1,943 |
| Immunoglobulin-like fold | 1,734 |
| Fibronectin, type III | 1,605 |
| Zinc finger C2H2-type/integrase DNA-binding domain | 1,543 |
| Ankyrin repeat | 1,517 |
| Cadherin | 1,217 |
| P-loop containing nucleoside triphosphate hydrolase | 1,106 |
| G protein-coupled receptor, rhodopsin-like | 1,037 |
| Src homology-3 domain | 856 |
| EGF-like, conserved site | 802 |
| PDZ domain | 788 |
| Immunoglobulin subtype | 744 |
| Protein kinase domain | 720 |
| Leucine-rich repeat, typical subtype | 720 |
| Sushi/SCR/CCP | 654 |
| Low-density lipoprotein (LDL) receptor class A repeat | 593 |
| Thrombospondin, type 1 repeat | 584 |
| Pleckstrin homology domain | 569 |
| EF-hand domain | 538 |
| Small GTPase superfamily | 494 |

The results of a gene ontology (GO) analysis assigned 11,695 unique proteins to 4,031 terms for biological processes, molecular functions, and cellular components. Within biological processes, metabolic (26%) and cellular processes (26%) were the most well-represented. Next, the majority of the proteins assigned to molecular functions were associated with binding (85%). Finally, within cellular components, cell (33%) and membrane (31%) proteins were the most well-represented (Fig. 3).

The most abundant conserved protein domain found in our data was the zinc finger C2H2 domain, followed by the immunoglobin-like fold and fibronectin-III. Zinc finger-associated conserved protein domains are 8% of all conserved domains, and 62.7% of those are associated with zinc finger C2H2 (Table 3).

The KEGG pathway analysis annotated 4,486 transcripts into 338 pathways, with 3.5 KEGG pathways per transcript on average. Of all the annotated sequences, a large portion (1,581, or 35.2% of 4,486) were related to metabolism, specifically of carbohydrates (332 sequences), lipids (247 sequences), and amino acids (291 sequences). Next, 855 sequences (19.1%) were involved in signal transduction. The most well-represented was the PI3K-Akt signaling pathway (ko04151; 224 sequences), which could be activated by IGF-1, followed by the MAPK signaling pathway (ko04010; 159 sequences). Finally, 483 sequences (10.8%) were associated with the immune system, including the T cell (ko04660) and B cell (ko04662) receptor signaling pathways.

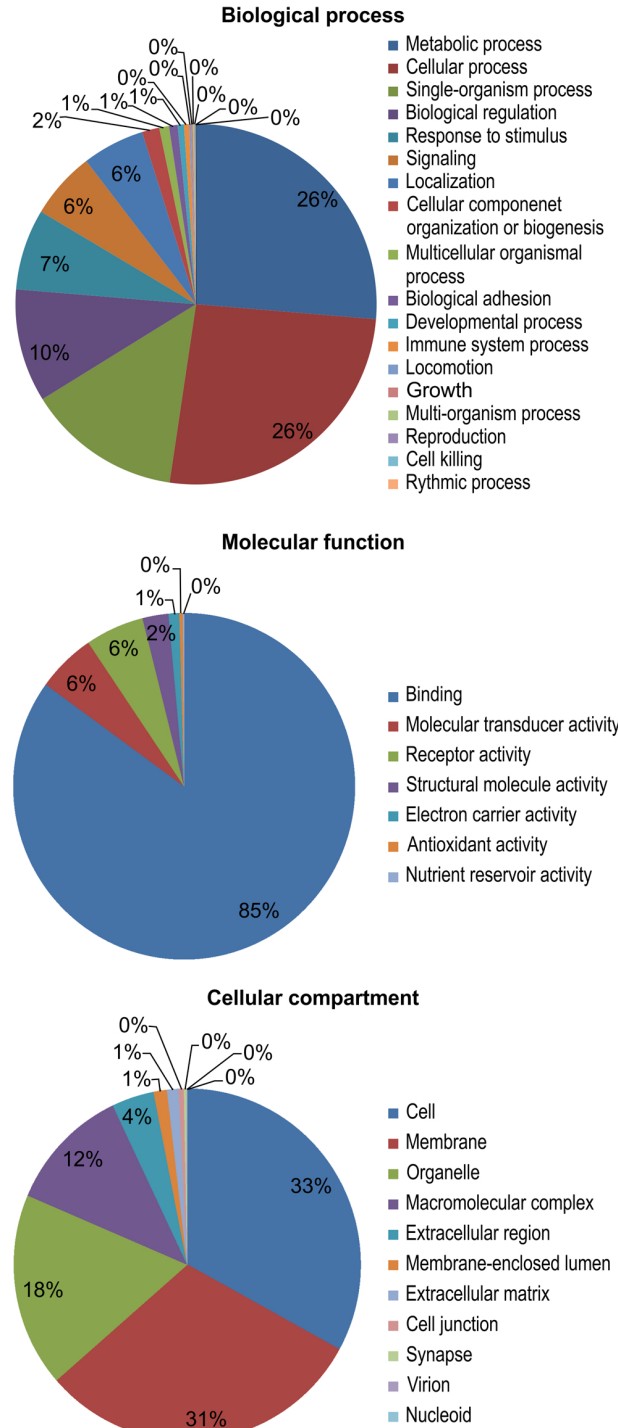

**Figure 3** Gene ontology (GO) functional categories of the *M. reevesii* assembly.
**Table 4** Sequence length and comparison of *tert* and TERT between *M. reevesii* and *C. picta, C. mydas, P. sinensis* and *H. sapiens.*

|            | M. reevesii | C. picta    | C. mydas    | P. sinensis | H. sapiens  |
|------------|-------------|-------------|-------------|-------------|-------------|
| *Tert* (nt) | 3,693       | 4,095(97%)  | 4,065(96%)  | 3,522(88%)  | 3,399(70%)  |
| TERT (aa)  | 1,230       | 1,352(96%)  | 1,354(93%)  | 1,173(81%)  | 1,132(60%)  |

### Longevity-related genes

Candidate genes related to longevity were screened out from the total pool of annotated genes. Specifically, fragments of *tert, tep1* and six (*trf1, trf2, tpp1, pot1, tin2* and *rap1*) shelterin proteins were found (File S2). Genes were translated referring to standard codon table and aligned with public database.

*Tert* gene, encoding the catalytic subunit of telomerase enzyme, has been drawn a lot of attention by biologists. We obtained a 3,817 bp of *tert* gene fragment including two conserve domains in 3″ terminal: RNA binding domain of telomerase (TRBD) and reverse transcriptase (RT) domain are essential for the gene function. The *tert* gene sequence length and identity were compared with other turtles and human (Table 4). Nucleotide conservation is interestingly higher than amino acid conservation, the similar results were also found in *Nothobranchius furzeri* (*Hartmann et al., 2009*).

FPKM values are approximately 10 of all aforementioned genes and genes that were closely related in function (File S3). The similar expression levels indicated that the candidate genes were accurately identified.

## DISCUSSION

### Profile of the *M. reevesii* transcriptome

More than 15-Gb high-quality data were generated with Illumina sequencing in this study. A total length of 16,478,555 bp (0.75% of 2.2 Gb) for 42,918 assembled transcripts were annotated that encoded for mRNA in *M. reevesii* transcriptome, functional annotation of the mRNAs provided fully information of transcripts and whole profile of the transcriptome that could be used in further study. The most abundant domain, C2H2 zinc finger, was consistent with the results found in mammalians and humans, the fact that the motif is the most prevalent and the largest sequence-specific DNA-binding protein family (*Lander et al., 2001*; *Tupler, Perini & Green, 2001*). The conservation and evolution of this domain could be further studied (*Englbrecht, Schoof & Böhm, 2004*).

Lastly, our functional annotation results also revealed a small amount of contaminated transcripts associated with viruses. While such data are negligible, they should nonetheless be excluded in future studies.

### The active expression of telomere-associated genes may contribute to turtle longevity

Telomere shortening is now considered the molecular clock that triggers cell and organismal senescence (*Harley & Goldstein, 1978*). To prevent premature shortening, telomere length is maintained by telomerase, previously considered to be inactive in human somatic

cells (*Kim et al., 1994*; *Shay & Bacchetti, 1997*). Using RNA-seq technology, we were able to find two coding genes of the telomerase complex—TERT and TEP1—in the *M. reevesii* transcriptome. Both genes are essential for proper telomerase function: transient expression of TERT has been found to reconstitutes telomerase activity (*Weinrich et al., 1997*), while TEP1 is a component of the ribonucleoprotein complex (*Poderycki et al., 2005*). While TEP1 has been closely linked to telomerase activity (*Nakayama et al., 1997*), other research has thrown doubt on this connection (*Uchida et al., 1999*). Thus, the exact function of TEP1 is a subject for further research. The presence of these two genes might implies that activated telomerase is present in turtle somatic tissues to protect the turtle from senescence.

We were able to detect all the reported six genes that encode shelterin proteins in our data. Expression of these genes could prevent the telomeres from shortening during cell division and thus were predicted having a significant effect to the longevity of this species.

## CONCLUSION

In this study, the transcriptome of the Chinese three-keeled pond turtle was sequenced using the Illumina Hiseq 2500 platform. A *de novo* assembly was then evaluated to uncover longevity-related candidate genes, mainly associated with telomere function, which offer a clue to the mechanisms behind turtle longevity. Future studies will incorporate RT-PCR and immunohistochemical techniques to test the gene expression levels and telomerase activity in different tissues. Finally, we believe the transcriptome data generated here can serve as a valuable resource for any investigations of turtle longevity and other notable characters in this order.

### Abbreviations

| | |
|---|---|
| *tert* | telomerase reverse transcriptase |
| *tep1* | telomerase protein component 1 |
| *trf1* | Telomeric Repeat Binding Factor 1 |
| *tpp1* | Tripeptidyl peptidase 1 |
| *rap1* | Ras-related protein 1 |
| *pot1* | Protection of telomeres 1 |
| *tin2* | TRF1-interacting nuclear factor 2 |
| FPKM | Fragments per Kilobase of transcript per Million mapped reads |

## ACKNOWLEDGEMENTS

We would like to thank Genergy for providing sequencing service and primary data analysis and Editage for providing editorial assistance. We also thank Xuming Zhou and PhD Xianzhao Kan for professional advice.

## Funding

The National Natural Science Foundation of China (NSFC, No. 31372198 and 30970351) and the Research Fund of the Key Laboratory of Biotic Environment and Ecological Safety of Anhui province funded the work. The funders had no role in study design, data collection and analysis, decision to publish, or preparation of the manuscript.

## Grant Disclosures

The following grant information was disclosed by the authors:
The National Natural Science Foundation of China: 31372198, 30970351.
Research Fund of the Key Laboratory of Biotic Environment: Ecological Safety of Anhui.

## Competing Interests

The authors declare there are no competing interests.

## Author Contributions

- Huazong Yin conceived and designed the experiments, performed the experiments, analyzed the data, contributed reagents/materials/analysis tools, wrote the paper, prepared figures and/or tables.
- Liuwang Nie conceived and designed the experiments, reviewed drafts of the paper.
- Feifei Zhao performed the experiments, contributed reagents/materials/analysis tools.
- Huaxing Zhou, Xianmei Dong, Huanhuan Zhang, Yuqin Wang, Qiong Shi and Jun Li assistance.
- Haifeng Li prepared figures and/or tables.

## Animal Ethics

The following information was supplied relating to ethical approvals (i.e., approving body and any reference numbers):

Procedures involving animals and their care were approved by the Animal Care and Use Committee of Anhui Normal University under approval number #20140111.

## DNA Deposition

The following information was supplied regarding the deposition of DNA sequences:
GenBank: SRX1469958.

## Data Availability

Figshare: https://figshare.com/s/aa9d1a8dc99ce8ace2bb
DOI: 10.6084/m9.figshare.2008326.

## Supplemental Information

Supplemental information for this article can be found online at http://dx.doi.org/10.7717/peerj.2062#supplemental-information.

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
