# Peer review of "De novo assembly and characterization of the Chinese three-keeled pond turtle (Mauremys reevesii) transcriptome: presence of longevity-related genes"

_PeerJ, doi:10.7717/peerj.2062_

## Round 0.1 · original submission · Major Revisions

I give credit to this transcriptome analyses of the Chinese three-keeled pond turtle. However based on available reviews, I recommend major revision and would ask that you provide point by point answers to the questions raised by the reviewers. I also recommend to avoid over-speculation in the discussion section.

·

Basic reporting

This paper reports the analysis of three Mauremys reevesii transcriptomes. As stated in the title, genes related to longevity are present in the dataset. The paper is honest and well written.

Experimental design

The experimental design is sound and coherent. Some details are missing (see general comments)

Validity of the findings

Findings and conclusions are supported by the analysis conducted by the authors.

Additional comments

Methods:

L113: FPKM is not an algorithm, more a unit to express corrected counts. RSEM use several algorithms to estimate gene expression, notably an EM algorithm (that give its name to the program) to deal with the multimap issue.
Also you will read that FPKM, as an absolute count of gene expression, have a bad reputation. See this article from Lior Pratcher the guy that came up with RPKM, FPKM and TPM:
https://liorpachter.wordpress.com/2013/11/02/stories-from-the-supplement/#comment-246

L114: Quality assessment and L120 Functional annotation overlap in their content. See Line 118: “top results were selected as the annotation of genes” is in the quality assessment part. It’s confusing. I think the quality assessment is more a “Comparison with turtles reference proteomes”.

Results:

L166: “Unique proteins” are not defined in the methods. Thus, table 1 is hard to understand since we only have been introduced to the unigene concept so far.
L167: I don’t understand why you state that 9.9% of positive unigene hits leads you to consider data acceptable and usable. I don’t mean that it’s not, I just don’t understand the rationale behind this affirmation. Maybe just remove it ?
L 208: You may want to provide a expression value for the IGF1R sequence to convince the reader that the deletion is not a lack of coverage. Or, even better, add to Figure 5 the reads mapping to the contigs so we clearly see the deletion.
**edit** I see it now. L216, you state that all genes have ~ 10 FPKM expression value. I suggest you give a table with the exact value for all longevity-related genes. Three samples should also give you the opportunity to give us the variance.

L 214: Nucleotide conservation higher than amino acid is really a weird result, not made clearer in the paper you cite (Hartmann 2009). Thus, I suspect an alignment issue. Can you please use a program that will transform your AA alignment in NT alignment ( like pal2nal or Macse ) and then compare those with the original NT alignment, for one species. Visual comparison could help you to provide an explanation.

Discussion:
First paragraph is a bold statement. Depth of coverage should be expressed as such (5X ? 10X ? )
L232: This (instead of the) […] for its conservation and evolution

L234: Filtering viral sequences from the raw reads is doable. Use, for example, deconseq.pl and a viral sequence database.

Figures and Table:
Results in figure 4 should be a table with number (sorted by abundance).

Reviewer 2 ·

Basic reporting

In this manuscript, Yin et al. perform a de novo assembly of the transcriptome of three tissues of the Chinese three-keeled pond turtle (Mauremys reevesii) and screen it in order to identify the presence of genes that in other species are known to be telomere-associated. I would like to see this published since it provides a useful resource in what is a small community: researchers studying turtles. However, they seem to fail in fact in that: I cannot find any accession number for the RNA-seq data in all the manuscript, and would like to remind the authors that they should upload their datasets to a public database like SRA from NCBI or ERA from EBI. Therefore, I cannot recommend this manuscript for publication without this requirement. Besides, without the public availability, the methods used and the re-analysis of the results cannot be replicated, an essential part of PeerJ standards, and of science in general.

Experimental design

This investigation has not been conducted rigorously and very easy experiments could have been done in order to validate its otherwise too speculative discussion. Besides, the Methods sections fails to describe in detail the procedures of their bioinformatics experiments, like parameters and their values used, as well as some software versions. See more details in the General Comments to the Authors.

Validity of the findings

Too speculative at the end of the Discussion section. I indicate it more in detaail in the General Comments for the Author.

Additional comments

I also have other suggestions that I feel would greatly improve the overall quality of the manuscript:

1) In the Methods sections, including the main text and the FileS1, the authors fail to provide the exact methodology. Importantly, some software versions are lacking, and most of the parameters and values used for them are not indicated. Please, to facilitate reproducibility, provide that information. For instance, is the Trinity version used for assembly the same as for functional annotation (trinityrnaseq_r20131110)? Importantly, the methods/software used for the clustering into Unigenes is completely lacking: “All assembled transcripts clustered into unigenes with only one transcript per locus, which reduces the complexity of the final assembly for subsequent analysis.” Please, state how this was performed: software, version, parameters.

2) Regarding their results, the authors find tert, tep1, igf1r and four (trf1, trf2, tpp1 and rap1) out of six shelterin proteins. I suggest that they blast some queries from other vertebrates were all shelterin proteins and other telomere-associated proteins in order to find transcripts that escape their functional annotation. Also, it would be very informative to to to those blasts also against the transcriptome/genomes of the other turtles (green sea turtle and Chinese soft-shelled turtle) in order to check the presence of other proteins, for instance the two shelterins not identified in Mauremys reevesii.

Without any other attempt to find those genes apart from the automatic functional annotation, the following sentence in L252-253 “while it is not clear why the remaining two were not detected, we believe this could be due to missing data and not an actual absence from the M. reevesii genome” significantly weakens the manuscript.

3) Since the work presented here is limited to the identification of the abovementioned genes, sentences like “The presence of these two genes implies that activated telomerase is present in turtle somatic tissues to protect the turtle from senescence” and “We were able to detect four of the six genes that encode shelterin proteins in our data. Expression of these genes could prevent the telomeres from shortening during cell division and thus had a significant effect to the longevity of this species” are too presumptuous and speculative and the results of this manuscript cannot support anything related to those asseverations. I suggest that the authors either delete these sentences or tone them down in order that it is clear that they are purely speculating at this point.

4) The stretched of a sequence stretch in the gene igf1r should be further studied. I am surprised that the authors draw any discussion around this, when they don’t try to reconstruct the whole CDS by other experiments. A clear one would be to synthesize cDNA from the RNA that has been used for the RNA-seq and try to obtain products from a PCR that uses primers designed at both identified fragments. I dare to say that I foresee that they will be able to find products that contain the missing sequence. Therefore, saying “assuming this was not the result of a sequencing error” and the discussion around it is too weak.

5) L154-155 - Note that N50 scaffold value is a measure of genome assembly, and not necessarily a good quality value for transcriptome assemblies.

6) L159 - What do you mean by “respectively”?

7) L226 – Change “A total of 16478555bp (0.75% of 2.2 Gb) transcripts” for “A total length of 16478555bp (0.75% of 2.2 Gb) for 459,911 assembled transcripts”. Correct number of transcripts if I am wrong.

8) L232 – “The domain could be further studied its conservation and evolution (Englbrecht et al. 2004)” makes no sense: “The conservation and evolution of this domain could be further studied (Englbrecht et al. 2004)” is what the authors mean?

---

## Round 0.2 · Minor Revisions

Both prior reviewers have commented on your revision. Please go through the points raised by the reviewers and make necessary changes before your manuscript is further considered.

·

Basic reporting

The authors made an thorough effort to follow the reviewers recommendation and amended their manuscript accordingly. They even made an additional experimentation that invalidates one of their previous finding and reported it honestly, thus demonstrating their scientific integrity.
I warmly advise the acceptance of this edited manuscript, provided a minor revision on the first paragraph of their Discussion is made.
L 241 to 246: I don't think it's pertinent to report an estimated coverage of the transcriptome. First, you have to make strong assumption on the size and coding fraction of the genome. Second, mRNAs are extremely variable in their level of expression, and thus the coverage of some can be very high, while others will remain undetected, whatever the depth of your sequencing is. Please report only the size of the generated datasets.
Or, if you're willing to give a sound estimation of your coverage, use the real distribution (raw coverage, not FPKM), provide a mean and variance and acknowledge unexpressed transcripts.

Experimental design

no comment

Validity of the findings

no comment

Additional comments

no comment

Reviewer 2 ·

Basic reporting

The authors have "repaired" the missing data by using complementary experiments to the RNA-seq/assembly. As I predicted, igf1r is actually complete, and thus they have now eliminated most speculation around the false deletion, and limited their manuscript to the description of the data for the other genes. Although this makes the manuscript inherently limited, it is now sounder and more robust. However, their revision of the manuscript is vague, the main text modifications have been limited to add a few words and delete some sentences and sections and they fail to honestly and accurately describe the real changes done. See my comments on Experimental Design about the two shelterins now found after my suggestion.

Experimental design

If the authors have found the two lacking shelterins after my suggestion, I wonder why in the manuscript these two genes appear like if they were found in their annotation? The authors should describe the real output of their results: that these two genes were lacking from their functional annotation, but that further specific BLASTs allowed finding them. Add the corresponding description in the M&Ms section as well, with the queries used.

Regarding the methods, let me insist in the "unigene" concept used. As far as this reviewer knows, and I might be wrong, Trinity does not work with unigenes, it works by clusters of "genes" (or what Trinity call genes) containing "isoforms". And although some softwares cluster the transcripts into unigenes, Trinity´s output is a multifasta file with all single transcripts for every Trinity "isoform" of each "gene" within each cluster. For the terminology used by Trinity, please visit here: https://github.com/trinityrnaseq/trinityrnaseq/wiki/Output-of-Trinity-Assembly. So, when the authors use RSEM to map back the reads to what they call "unigenes", I guess they have used the script "align_and_estimate_abundance.pl" found in the Trinity util/ folder within the Trinity package. If so, I suggest that they erase L116-121 and just mention that they used the Trinity package on the final Trinity output. Also, correct elsewhere where you menteon unigenes if you have used the Trinity output (transcripts instead of unigenes). If the authors have selected a single representative transcript per gene (e.g., the most abundant, the longest....) by other means, please state so and describe it properly in the methods section.

Last, indicate in the manuscript, where needed and not already stated, that you have used default parameter values, by adding for example "with default parameters".

Validity of the findings

The findings and methods are sound. But still, the authors need to change the manuscript to adjust it to what they have really done.

Additional comments

- In the last sentence of the abstract, delete "In addition".
- Line 99: correct CDNA to cDNA

---

## Round 0.3 · accepted · Accept

After re-review, the reviewers agreed that no further amendments are required in this manuscript. We are pleased to inform you that this manuscript is accepted now.

·

Basic reporting

I believe that, after those rounds of review, the authors have amended and improved their manuscript enough to warrant its publication in PeerJ

Experimental design

no comment

Validity of the findings

no comment

Additional comments

no comment.

Reviewer 2 ·

Basic reporting

No comments

Experimental design

No comments

Validity of the findings

No comments

Additional comments

I have no further concerns regarding this manuscript.